# Constructing and influencing perceived authenticity in science communication: Experimenting with narrative

**Lise Saffran**[1☯], **Sisi Hu**[2☯]*, **Amanda Hinnant**[2☯], **Laura D. Scherer**[3‡], **Susan C. Nagel**[4‡]

**1** Department of Public Health, University of Missouri, Columbia, Missouri, United States of America,
**2** School of Journalism, University of Missouri, Columbia, Missouri, United States of America, **3** School of Medicine, University of Colorado, Denver, Colorado, United States of America, **4** Obstetrics, Gynecology and Women's Health, University of Missouri, Columbia, Missouri, United States of America

☯ These authors contributed equally to this work.
‡ These authors also contributed equally to this work.
* sisihu@mail.missouri.edu

**Data Availability Statement:** All data files are available from the Open Science Framework (OSF) database (accession URL: https://osf.io/7wcg8/?

## Abstract

This study develops a measure of perceived authenticity in science communication and then explores communication strategies to improve the perceived authenticity of a scientific message. The findings are consistent with literature around trust and credibility, but indicate that authenticity—the perception that the scientist is a unique individual with qualities *beyond* institutional affiliations or a role in the production of the research—may add a potentially important dimension to accepted categories of integrity and benevolence.

## Introduction

The traditional approach of science communicators, which seeks to fill in a knowledge deficit on the part of the public, is ineffective in a changing media and political environment according to the National Academies of Sciences, Engineering and Medicine's *Communicating Science Effectively: A Research Agenda* (NAS report) [1]. The NAS report further argues the bidirectional aspects of science communication. It underscores the fact that even as the scientist is evaluating their audience and adapting messages accordingly, the audience is also gauging the scientist. The NAS report identifies trust as being caused by the audience's perception of the communicator's integrity, dependability, and competence [1]. Examining trustworthiness in the context of organizational behavior, Mayer, Davis, and Schoorman [2] identify these qualities as expertise, integrity and benevolence, consistent with a wide variety of research into credibility and trustworthiness [3–5]. In each case, trust is caused by a collection of factors that includes judgments about the communicator's character and intentions, as well as their ability and knowledge.

The literature in science communication broadly supports the idea that trust is affected by audience perceptions that a potential trustee is motivated by good intentions and possessed of good character. The NAS report [1] calls these characteristics integrity and dependability, Hendriks, Kienhues, and Bromme [3] refer to them as integrity and benevolence. These

view_only=5831825b4240466a806fbc0a0f1
cdf16).

**Funding:** The study was funded by the Research
Council Grants at the Office of Research and
Economic Development in the University of
Missouri within the project titled, "Perceived
Authenticity as a Cause of Trust in Science
Communication." Research Council Grant Number:
URC-18-045. URL: https://research.missouri.edu/
internal/researchcouncil. Funding received: LS. The
funder had no role in study design, data collection
and analysis, decision to publish, or preparation of
the manuscript.

**Competing interests:** The authors have declared
that no competing interests exist.

elements of trustworthiness associated with benevolence and integrity overlap substantially
with the qualities identified by Fiske and colleagues [6] as contributing to perceptions of
warmth, a quality that combined with judgments about competence form universal dimen-
sions of social cognition. Research by Schoorman, Mayer, and Davis [7] acknowledges that the
establishment of trust is not exclusively a cognitive activity and that affective responses to the
communicator and indeed emotional states have the power to influence trust.

The aim of this study is to examine how the concept of authenticity might offer a compre-
hensive framework for the causes of trust identified in the literature as *integrity* and *benevo-
lence*. Based on a review of authenticity literature in social psychology, education, marketing
and communication, public health communication, organizational behavior and narrative
studies, our a priori definition of perceived authenticity is this: *the belief by the message receiver
that the communicator is a human being with their own history, values, and point of view and
that the message they are communicating is in accordance with those values*. While there is a sig-
nificant body of interdisciplinary literature that examines authenticity with regard to self [8–
9], intimate partner relationships [10] and marketing/branding [11–12], the nature of authen-
ticity in science communication has, to our knowledge, not been examined.

As there is currently no useful measure of perceived authenticity in the context of science
communication, a primary goal of this research is to develop a valid measurement tool. The
instrument tested here was designed to be expansive in measuring the resemblance of authen-
ticity to *benevolence* and *integrity* in order to capture qualities that may not be entirely reflected
in those two widely accepted causes of trust, including reciprocity and warmth. The definition
used in this research intentionally excludes the concept of expertise. In the research of Fiske
and Dupree [13] warmth is judged before competence, and additional research suggests that
judgments of expertise are most vulnerable to motivated reasoning, particularly in politicized
contexts [14–15].

In sum, this research first explores perceived authenticity as a method of conceptualizing
the perception that the scientist is a unique individual with qualities *beyond* institutional affili-
ations or a role in the production of the research, then develops and tests an instrument for
measuring the perception of authenticity and, finally, explores how authenticity might be com-
municated, and manipulated, through narrative in a non-politicized context.

## Conceptions of authenticity

Ancient Greek philosophers, including Socrates, describe an ideal combination of self-knowl-
edge, benevolence, and ethical behavior that is echoed in contemporary definitions of authen-
ticity [8]. The presence of a human being who possesses self-knowledge, clarity in their
convictions, an individual set of values, and a commitment to acting in accordance with those
values lies at the heart of much of the definitions of authenticity in social psychology literature
[16]. Much of the psychological literature examining authenticity focuses on an individual's
own self-concept and behavior, for example, the authenticity inventory built by Kernis and
Goldman around a multi-component conceptualization that includes self-understanding,
unbiased processing, behavior that is consistent with one's values, and places value on others
perceiving the "real you" [8]. An authentic speaker is one who is "[S]incere, innocent, original,
genuine, and unaffected," [17] as well as transparent with regard to intentions, point of view
and values [18] and who is willing to be personally associated with the message being conveyed
[11].

The literature examining authenticity in the context of marketing and consumer behavior
examines authenticity as a quality which is both conveyed and perceived [11, 19]. In organiza-
tional and management literature, the definition of authenticity often emphasizes the

relational aspect [20–21], an emphasis which is also found in literature examining authenticity in the context of education [22–23].

The education literature is particularly relevant in this case, because perceived authenticity offers the message receiver an opportunity to recognize that the communicator is a unique individual with a set of values, a history, and intentions, as they themselves are. Johnson and LaBelle's [22] research into teacher authenticity acknowledges the challenge of conveying authenticity in the context of a power imbalance, a context which offers the possibility that the communicator may be perceived as "looking down" on the message receiver [22]. The potential of authenticity to capture the dimensions of trust that focus on the communicator's "having a personal stake in the issue," their similarity to the receiver" and their "willingness to admit uncertainty" [24] have particular relevance in science communication, where the communicator is possessed of presumed expertise and advanced knowledge. It is with regard to the reciprocal summing up by communicator and message receiver, particularly in an inherently unbalanced exchange (what Petraglia refers to in public health communication as dialogue [25]) that perceived authenticity offers areas of potential difference from traditional conceptions of *integrity* and *benevolence*.

## Communicating authenticity in science

In developing science communication messages with the potential to *convey* what Rabinovich et al. describe as "transparency about the intentions, point of view, and values of the communicator [18], we turned to the literature in health humanities, public health communication and literary studies. Literature in psychology, public health communication and narrative suggests that narrative strategies, particularly those that focus convincingly on individuals with complicated inner lives, have the potential to increase empathy across cultural divides and diminish message resistance [26–27]. Further, strategies that encourage people to focus on the unique qualities of individuals may be effective in reducing bias toward the group those individuals represent, according to social psychology research into individuation [28]. In this framework, authentic communication is communication in which an individual person is perceived to be discernable and, while not personally known, is nonetheless recognizable enough to elicit a personal response on the part of the audience.

Understanding how authenticity might be constructed and measured in science communication is particularly important in an environment where an increasing number of people obtain science information through the internet [3]. For lay people, assessments of credibility and trustworthiness are complicated by the inconsistent nature of information available online about the communicators of scientific information: information can be incomplete, misleading and difficult to source [3]. Further, as scientists increasingly incorporate narrative into their communication efforts, there may be unintended consequences to attempting authentic communication without having studied the qualities of effectively authentic messages. There is evidence that *inauthentically* communicating even a true message may diminish trust [29].

In determining the credibility of a source, people often rate *perceived* expertise—related to having common interests with the speaker and perceptions of trustworthiness—over *actual* expertise, for example an advanced degree in the subject [30]. In deciding about these factors, the audience may be assessing the organization that the scientist represents, the medium for communication, relevant political, social or economic considerations and potential threats to their identity and values [31–32]. People reject scientific information that contradicts their own beliefs and seek out information that confirms their beliefs in the process called motivated reasoning [33]. While some science topics—climate change, vaccination—are clearly linked to social and political identity in public discourse [34–35], the politicization of science topics

generally is in flux and subject to a variety of trends [36]. A style of communication that relies on objectivity and traditional scientific authority is undermined by motivated reasoning, particularly when topics are related to political ideology [37].

If authenticity in communication "enables individuals to understand, emotionally as well as cognitively, how information can relate to their everyday existence," [25], then a sense that scientists, too, are grappling with questions of uncertainty, particularly around value-laden issues, offers a potential avenue of connection and even empathy.

## Materials and methods

### Hypotheses and research questions

The purposes of this study are first to develop a measure of perceived authenticity in science communication and then to explore which communication strategies can improve the perceived authenticity of a scientist. The first part of this research inquiry asks:

**RQ1:** What is a valid measure for perceived authenticity in science communication?

Based on previous research, four strategies alternately employed in the scientific brief were hypothesized to enhance the authenticity of a message: 1) first-person narrative style alone; 2) presenting forward-looking fallibility; 3) presenting the researcher's origin of interest; 4) presenting backward-looking fallibility. All experimental messages were presented in the first person in order to explicitly connect the message with the individual communicator without an intermediary.

It is hypothesized that a scientific brief with any of those four strategies will increase the perceived authenticity of the scientist when compared to the perceived authenticity of a scientific brief without any of those four strategies.

**H1:** First-person narrative style in a scientific brief will be perceived as more authentic than the control message.

The narrative components in the message conditions involve, in each case, conveying that the science communicator is an individual with a specific point of view, values and interior life. The first experimental message condition employs the first person narrative voice to convey, in the simplest way, that a single individual is communicating to the audience. Hypothesis 1 thus provides a useful basis for further message design. Each of the other three strategies are added to the first-person narrative style. Therefore, the additional hypotheses are:

**H2:** First-person narrative style combined with presenting forward-looking fallibility in the scientific brief will be perceived as more authentic than the control message.

**H3:** First-person narrative style combined with presenting the researcher's origin of interest in the scientific brief will be perceived as more authentic than the control message.

**H4:** First-person narrative style combined with presenting backward-looking fallibility in the scientific brief will be perceived as more authentic than the control message.

**RQ2:** Which narrative strategy or strategy-combinations in a scientific brief are perceived as more authentic?

### Design and stimuli

This study was a between-subjects online experiment with a U.S. sample that manipulated message strategies within a vignette that was written as a brief report of scientific research. The

chosen topic was non-controversial and non-political (the domestication of plants in agriculture) in order to avoid the influence of other social and political factors.

The elements of narrative used to construct this study's 5 narrative message conditions are theorized to promote empathy through identification [38]. The study design utilizes first person point of view (Conditions 2–5), which explicitly ties the message to the messenger [39–40]. Thus, the 5 experimental conditions designate a single communicator with whom readers could identify: a scientist with a point of view, history, and individual concerns. The intention of this study to measure authenticity directly, as opposed to isolating and measuring factors associated with authenticity (such as uncertainty or vulnerability), led to the adoption of the first person in all experimental message conditions for the following reason: In this study perceived authenticity relates to the perception that the communicator is an individual person and responsible, as an individual, for the message being communicated. Presenting qualities such as vulnerability in the third person would introduce yet another person into the communication exchange—the narrator. Including a third person could risk undermining the connection between the message and communicator.

Perceived authenticity is also associated with transparency, honesty and openness [11], verisimilitude [41], voice [42] and the presence of an unruly narrative [40]. These indicate a tendency by readers to consider narratives of contestation and opposition more authentic, for example content that is at odds with what audience members might expect the communicator's perspective to be, such as a scientist who is willing to admit a previous mistake to their audience or one who is open to the possibility that future discoveries may undermine current findings (Conditions 3 and 5). The NAS report also connects trust to "the willingness of both parties to take risks and be vulnerable" [1], which in the context of science communication could also take the form of a scientist's admitting previous errors (backward-looking fallibility) or highlighting uncertainty in their work (forward-looking fallibility). In our definition of forward-looking fallibility, what the communicator is conveying is a specific kind of scientific uncertainty: the idea that future discoveries might prompt a reevaluation of present findings. The origin of the scientist's interest in the subject under study (Condition 4) relates to the importance of provenance in much authenticity literature, the idea that an object, or message, has an organic, as opposed to fabricated, origin [11]. See S1 Appendix for the scientific briefs and more detail on each message condition.

## Participants and procedure

Perceived authenticity is fundamentally a type of attitude, and prior research has shown that effect sizes for attitude change interventions are typically small to moderate [43]. A priori power analysis conducted using G*Power showed that 40 participants are needed for each cell in order to detect a medium effect (f = .25) for a *F*-test with a between-subjects factor to have a power of .80 at an alpha level of .05. Considering the attention issue of the online panel study and the possible outliers, a total of 500 U.S. based participants were recruited for this experiment. Amazon Mechanical Turk (MTurk) was used to recruit the participants. MTurk is a web-based crowdsourcing survey platform that is widely used to recruit participants for social science research and shown to generate reliable and valid data [44–45].

After participants consented to the study, they were instructed to read a passage from a plant science researcher describing new findings on plant domestication that offered dating ancient corn cobs as an example of research. Next, the participants were randomly assigned to one of the five scientific brief conditions. After their exposure to the passage, participants answered an attention check question which required them to choose among four options (bananas, dogs, corn cobs, chickens) in response to the question, "The passage you just read

talks about__". They then completed a 20-item authenticity questionnaire. Finally, participants answered demographic questions, were thanked for their participation, and were compensated $0.5. The average amount of time spent on the study was 3.94 minutes ($SD$ = 2.66, range = 1–25).

Demographic questions included gender, age, race, education, and political identification. Education was measured by asking participants to select their highest level of degree on a 6-point scale, from less than high school degree (1) to a graduate degree (6). Political identification was measured on a 7-point scale, from strong Democrat (1), to no preference (4), to strong Republican (7). See S1 Appendix for the full list of survey questions.

## Dependent variable scale development

**Authenticity.** Because there is no established scale for perceived authenticity, we developed 19 questions adapted from research into dispositional authenticity by Kernis and Goldman [8] and Wood et al. [9] and with reference to the research of Johnson and LaBelle [22] on teacher authenticity. For example, the instrument developed by Wood et al. [9] measures authenticity through agreement on a 5-point scale with statements such as "I always stand by what I believe in." These items were adapted for the current study to refer to the science communicator, for example, "This researcher believes in the findings he/she is presenting." When developing our measurement tool we took particular note of efforts to measure authenticity in contexts where the target of the assessment occupies a position of relative power. This approach acknowledges the imbalance inherent in the exchange; the communicator is in fact an expert and thus in a privileged position—certainly with regard to knowledge but also, potentially with regard to education and status—vis a vis the audience. This is in contrast to questions of authenticity that arise in other contexts, such as intimate partner relationships where the perceptions are toward someone of equal status or in marketing, where the audience holds a presumed power over the communicator (to buy or not to buy). Thus, Johnson and LaBelle's research on teacher authenticity formed the basis for a series of additional measures in this study related to whether or not the communicator is perceived to "look down" on readers and whether she desires to hide behind a "smoke screen of professionalism" [22].

The 19 questions were drafted to measure five aspects of authenticity drawn from the literature that reflected both emotional and cognitive aspects. The measures included these concept groupings: openness, unbiased processing, self-awareness, relevance of the research for the audience, and the researcher's passion for the research. For the purpose of this study, the category of "relevance" was added to reflect both the perceived inherent value of the information being presented as well as the communicator's good faith efforts to make sure that the audience can relate to the information in a personal way. The survey included questions such as, *How likely is this researcher to hide their true thoughts, feelings and doubts behind their role as a researcher*? (Reverse coding) (1 = very unlikely; 5 = very likely). Seven out of the 19 questions were designed as reverse wording questions. We acknowledge that reverse-wording strategy does not reduce response bias that results from participants' acquiescence, inattention, or confusion on the items [46]. Using reverse-worded items in this study was not for those purposes, however, but to improve interpretation of the items. Those seven questions flowed better and were therefore easier to understand when phrased in a reverse direction. Therefore, we decided to keep those reverse-worded questions in the questionnaire and later reverse coded them in the statistical analysis. A final question (20) "*How authentic do you perceive this scientist to be*? (1 = least authentic; 5 = most authentic) was included as a validation item. All authenticity measures used a 1–5 Likert scale.

## Results and discussion

### Sample overview

Participants who failed the attention check question (68 participants) were removed from analyses, leaving a total analytic sample size of $N = 432$ ($M_{age} = 35.06$ years, $SD = 11.20$, range = 18–73). The sample included 235 males (54.4%) and 197 females (45.6%), and participants were primarily White/Caucasian ($n = 337$, 78%), followed by Black/African American ($n = 39$, 9%), Asian ($n = 39$, 9%), and American Indian or Alaska Native ($n = 8$, 1.9%). With regard to education level, nearly half of the participants had bachelor's degrees ($n = 200$, 46.3%), 38.2% ($n = 200$) had less than a bachelor's degree, and 15.5% of participants ($n = 67$) had a graduate degree. With regard to political identification, nearly half of the participants identified as Democrat ($n = 202$, 46.8%), 155 (35.8%) identified as Republicans, and 75 (17.4%) indicated no preference. Among the 432 U.S. adults, 92 saw the control message, 83 saw the first-person condition, 83 saw the condition referencing uncertainty with regard to the findings, 87 saw the condition referencing the origin of the science communicator's interest in the subject under study, and 87 saw the condition referencing earlier mistakes in the scientist's interpretation of the data/analysis.

### Factor analysis

Research question 1 seeks to develop a valid scale of authenticity in science communication. Two exploratory factor analyses (EFA) were conducted to answer this research question. The results revealed that 11 of the 19 items were coherent and consistent measures of two factors of authenticity. In the initial EFA, all 19 items were expected to be related to authenticity. This

**Table 1. Structure matrix for the two factors solution.**

| Item | Factor 1 | Factor 2 |
|---|---|---|
| *Factor 1*: Connection | | |
| How respectful do you think this researcher is of their audience? | .703 | |
| How passionate do you think this researcher is about their area of research? | .682 | |
| How well do you think this researcher understands why he/she does the things he/she does? | .646 | |
| How knowledgeable do you think this researcher is about their area of study? | .625 | |
| How well do you think this researcher understands his or her own biases, motivations and influences? | .597 | |
| How important is it to this researcher that you understand their findings? | .548 | |
| *Factor 2*: Integrity | | |
| How likely is it that this researcher would be swayed in their research for personal gain? | | .782 |
| How influenced do you think this researcher is by factors outside the study (e.g. funders, employers or colleagues)? | | .700 |
| To what extent is this researcher the type of person who would use their role as a scientist to place themselves above other people? | | .656 |
| How likely is this researcher to hide their true thoughts, feelings and doubts behind their role as a researcher? | | .625 |
| How strongly does this researcher allow him or herself to be influenced by other people? | | .621 |
| Eigenvalue | 3.65 | 1.12 |
| Reliability | .80 | .81 |
| Variance explained | 33.16 | 10.19 |

$N = 432$. Extraction Method: Principal Axis Factoring. Rotation Method: Direct Oblimin

EFA used principal axis factoring with direct oblimin rotation because it is generally expected in social science that the factors would be correlated [47]. The criterion for factor extraction was that the eigenvalue be larger than 1. Three factors were extracted. Items were assigned to a particular factor if their primary loadings were greater than .5, which is a desirable cutoff point indicated in previous literature [48]. After examining the loadings of the three factors, seven items loaded under the first factor, eight items loaded under the second factor, whereas no item loaded greater than .5 under the third factor. After this, we re-examined each item based on its loading and logic, and 11 items from the first two factors were kept. The eight items with inadequate loadings shared some conceptual characteristics, mainly they often asked participants to engage in more complex thought thereby leading to confusion and/or they asked participants to speculate to a greater degree than other questions.

To re-test and confirm the selection, a second EFA was conducted. In the second EFA, 11 items were factored by principal axis factoring with direct oblimin rotation. Two factors emerged. Items which loaded under the first factor substantially resembled descriptions of *integrity* in the broader literature, reflecting unbiased processing and transparency/honesty. Items loading under the second factor included those that resembled *benevolence* in the literature (respectful of audience) but also included perceived qualities in the researcher that had emotional dimensions (passion for the research, a desire to be understood) as well as self-understanding and knowledge of the subject. We labeled the second factor *connection* to reflect the connection of the researcher to three dimensions: self, research, audience.

Among the 11 items, six were loaded under the factor *connection* and the other five were loaded under the factor *integrity*. The factors *connection* and *integrity* cumulatively accounted for 43.35% of the variance among the items (see Table 1). Specifically, the *connection* factor had an eigenvalue of 3.65, accounting for 33.16% of the variance; its loaded items have a Cronbach's alpha of .80. The *integrity* factor had an eigenvalue of 1.12, accounting for 10.19% of the variance; its loaded items have a Cronbach's alpha of .81. The correlation between these two factors was .494, which indicated that the correlation was not so high as to suggest these two factors were measuring the same construct [49]. According to the factor analysis, two dependent variables, *connection* and *integrity*, were formed by calculating the mean score of all items loaded under each factor, respectively; both variables scored on a scale of 1 (least) to 5 (most).

We subsequently tested the convergent validity of the factors against participants' self-reported perceived authenticity through a linear regression with the factors *connection* and *integrity* as independent variables, and the self-reported perceived authenticity as the dependent variable. The result showed both the *connection* and *integrity* factors were significantly related to perceived authenticity, with *connection* ($B = .65$, $SE = .05$, $p < .001$, $\Delta R^2 = .37$) accounting for a larger portion of authenticity judgments than *integrity* ($B = .17$, $SE = .05$, $p < .001$, $\Delta R^2 = .03$). The total variance in authenticity explained by these two factors (41%) was similar to the factor analysis result (43%). Therefore, we consider this two-dimensional measurement to be a valid measure of perceived authenticity for hypothesis testing.

## Hypotheses testing and second research question

A multivariate analysis of covariance (MANCOVA) determined the effect of the five different message conditions on the two factors of authenticity (i.e., connection and integrity), after controlling for participants' education and political ideology. The five message conditions were set as an independent variable, education, and political ideology were set as covariates, and the two factors (connection and integrity) were dependent variables. Preliminary assumptions checking revealed that data was normally distributed, observed from normal Q-Q plots; there were linear relationships between covariates and dependent variables for each condition,

**Table 2. Means, adjusted means, standard deviations, and standard errors for connection and integrity for different message conditions (N = 432).**

| Condition | Authenticity | | | | |
| | Connection | | Integrity | | |
| | M (SD) | M_{adj}(SE) | M (SD) | M_{adj}(SE) | n |
|---|---|---|---|---|---|
| Condition 1: conventional academic format (control) | 3.93(.72) | 3.93(.06) | 3.12(.77) | 3.13(.09) | 92 |
| Condition 2: first person | 4.19(.54) | 4.19(.07) | 3.45(.95) | 3.44(.09) | 83 |
| Condition 3: first person and referencing forward-looking fallibility | 4.09(.59) | 4.07(.07) | 3.22(.92) | 3.16(.09) | 83 |
| Condition 4: first person and referencing the origin of the science communicator's interest in the subject under study | 4.20(.56) | 4.22(.07) | 3.45(.91) | 3.49(.09) | 87 |
| Condition 5: first person and referencing backward-looking fallibility | 4.03(.64) | 4.04(.07) | 3.41(.89) | 3.45(.09) | 87 |

as assessed by scatterplot; there was homogeneity of regression slopes, as assessed by the interaction term between education and condition, $F(8, 832) = .77$, $p = .63$, and political identification and condition, $F(8, 832) = .40$, $p = .92$; there was homogeneity of variance and covariance, as assessed by Box's M test ($p = .002$); there was one univariate outlier for connection, as assessed by standardized residuals greater than ± 3 standard deviations, and two multivariate outliers, assessed Mahalanobis distance values greater than a cut-off point of 13.82, in which the one univariate outlier was also one of the multivariate outliers. These two outliers were kept because the results were not substantially affected after comparing the results of the MANCOVA with and without the outliers. Table 2 shows the means, adjusted means, standard deviations, and standard errors for connection and integrity for each message condition, in which means and adjusted means were not dissimilar.

There was a statistically significant difference between message conditions on the combined dependent variables (i.e., connection and integrity) after controlling for education and political identification, $F(8, 848) = 2.93$, $p = .003$; Wilks' $\Lambda = .95$; partial $\eta^2 = .03$. Follow up univariate one-way ANCOVAs were performed with a Bonferroni adjustment. There were statistically significant differences in adjusted means for both connection ($F(4, 425) = 3.37$, $p = .01$, partial $\eta^2 = .03$) and integrity ($F(4, 425) = 3.71$, $p = .006$, partial $\eta^2 = .03$) among different message conditions. Pairwise comparisons with a Bonferroni adjustment were made for both connection and integrity factors of authenticity (see Table 3). It should be noted that pairwise comparisons were conducted in SPSS, in which p-value reported will be 1.000 when the product of unadjusted p-value and the number of comparisons exceeds 1.

**Hypothesis 1** proposed that the first-person narrative style in a scientific brief (message condition 2) will be perceived as more authentic than the conventional style control message. This hypothesis was partially supported by the data. Participants who saw the scientific brief using the first-person narrative style rated statistically significantly higher on connection ($M_{dif} = .26$, $p = .045$) than those who saw the control message, but not significantly higher on integrity ($M_{dif} = .31$, $p = .15$).

**Table 3. Pairwise contrasts for adjusted means for two authenticity factors for each message condition.**

| Authenticity | Difference in adjusted means (95% CI) | | | |
| | Condition 2 vs. Condition 1 | Condition 3 vs. Condition 1 | Condition 4 vs. Condition 1 | Condition 5 vs. Condition 1 |
|---|---|---|---|---|
| **Connection** | .26 (.00, .52)* | .14 (-.12, .39) | .29 (.03, .54)* | .11 (-.15, .36) |
| **Integrity** | .31 (-.05, .67) | .03 (-.33, .39) | .36 (.01, .71)* | .32 (-.04, .67) |

* = statistically significant difference ($p < .05$) based on Bonferroni adjustment; 95% confidence interval (CI) is simultaneous confidence interval based on Bonferroni adjustment; Condition 1 = Scientific brief written in conventional academic format (control); Condition 2 = Scientific brief written in the first person; Condition 3 = Scientific brief written in first person and referencing forward-looking fallibility; Condition 4 = Scientific brief written in first person and referencing the origin of the science communicator's interest in the subject under study; Condition 5 = Scientific brief written in first person and referencing backward-looking fallibility.

**Hypothesis 2** proposed that the first-person narrative style combined with presenting forward-looking fallibility in the scientific brief (message condition 3) will be perceived as more authentic than the control message. This hypothesis was not supported by the data. Compared with those who saw the control message, participants who saw the scientific brief using the first-person style combined with presenting uncertainty with regard to the findings did not rate statistically significantly different on either connection ($M_{dif}$ = .14, $p$ = 1) or integrity ($M_{dif}$ = .03, $p$ = 1) factors of authenticity.

**Hypothesis 3** proposed that the first-person narrative style combined with presenting the researcher's origin of interest in the scientific brief (message condition 4) will be perceived as more authentic than the control message. This hypothesis was fully supported. Participants who saw the scientific brief using the first-person narrative style combined with presenting the researcher's origin of interest rated statistically significantly higher on both connection ($M_{dif}$ = .29, $p$ = .02) and integrity ($M_{dif}$ = .36, $p$ = .04) than those who saw the control message.

**Hypothesis 4** proposed that the first-person narrative style combined with backward-looking fallibility in the scientific brief (message condition 5) will be perceived as more authentic than the control message. This hypothesis was not supported by the data. Participants who saw the scientific brief in this condition did not rate statistically significantly higher on either connection ($M_{dif}$ = .11, $p$ = 1) and integrity ($M_{dif}$ = .32, $p$ = .12) of authenticity.

**Research question 2** investigated which strategy or strategy combinations in a scientific brief can make the science communicator be perceived as more authentic. Pairwise comparisons with a Bonferroni adjustment in previous MANCOVA showed the message condition 4 (i.e., the scientific brief writing in the first-person narrative style combined with presenting the researcher's origin of interest) did not elicit significantly higher scores for either connection or integrity factors of authenticity when compared to condition 2 (i.e., the scientific brief writing in the first-person narrative style only), condition 3 (i.e., the scientific brief writing in the first-person narrative style combined with forward-looking fallibility), or condition 5 (i.e., the scientific brief written in the first-person narrative style combined with backward-looking fallibility). However, according to the findings from H1 to H4, message condition 4 was the only message which elicited significantly higher scores on both connection and integrity of authenticity when compared to the control message. Therefore, it is appropriate to conclude that message condition 4, the combination of the first-person narrator and presenting the researcher's origin of interest in the scientific message, were perceived as more authentic.

## Exploratory analyses

The MANCOVA result above showed that the covariates, education ($F$(2, 424) = 6.76, $p$ = .001, Wilks' $\Lambda$ = .97, partial $\eta^2$ = .03) and political identification ($F$(2, 424) = 24.18, $p < .001$, Wilks' $\Lambda$ = .90; partial $\eta^2$ = .10) were significantly associated with the combined dependent variables. Though not hypothesized, exploratory analyses were conducted to determine whether education and political identification interacted with message conditions to affect the perceived authenticity of the scientist. First, two interaction terms (i.e., between education and the condition variable, and between political identification and the condition variable) were added into the above MANCOVA models; however, there were no statistically significant interaction effects for either the interaction between education and conditions ($F$(8, 832) = .77, $p$ = .63, Wilks' $\Lambda$ = .985) or between political identification and conditions ($F$(8, 832) = .40, $p$ = .92, Wilks' $\Lambda$ = .992).

To further test the additional unique effects of education and political identification on authenticity after accounting for the effects of message conditions, two hierarchical linear regressions were conducted. Four dummy-coded conditions, education, and political

**Table 4. Result of regression analyses for predictors of connection and integrity (n = 432).**

| Predictor | Connection | | | Integrity | | |
|---|---|---|---|---|---|---|
| | $\Delta R^2$ | B (S.E.) | ß | $\Delta R^2$ | B (S.E.) | ß |
| *Condition Variables* | .03* | | | .02* | | |
| Condition 2 | | .26 (.09) | .17** | | .31 (.13) | .14* |
| Condition 3 | | .14 (.09) | .09 | | .03 (.13) | .01 |
| Condition 4 | | .29 (.09) | .19** | | .36 (.13) | .16** |
| Condition 5 | | .11 (.09) | .07 | | .32 (.13) | .14* |
| *Interested Variables* | .04*** | | | .12*** | | |
| Education | | -.08 (.02) | -.15** | | -.09 (.03) | -.13** |
| Political identification | | -.04 (.02) | -.12* | | -.16 (.02) | -.32*** |
| Total $R^2$ | .07 | | | .14 | | |
| F Statistic | 5.05*** | | | 11.83*** | | |

Condition 2 to 5 were all dummy-coded, with the control condition 1 as the reference group.

*$p < .05$.

**$p < .01$.

***$p < .001$.

identification were the independent variables, and connection and integrity were the dependent variables, respectively. Table 4 displays the results of the hierarchical regressions for the both connection and integrity factors. For the prediction of the connection factor of authenticity, the model accounts for a significant portion of the dependent variable, $F(6, 425) = 5.05$, $R^2 = .07$, $p < .001$. The unique and statistically significant contribution of the education and political identification was 4%. Participants with higher education level ($B = -.08$, $SE = .02$, $p = .001$) and who self-identified more as Republican ($B = -.04$, $SE = .02$, $p = .01$) rated the connection factor of authenticity statistically significantly lower. For the prediction of the integrity factor, the model also accounts for a significant portion of the dependent variable, $F(6, 425) = 11.83$, $R^2 = .14$, $p < .001$. The unique and statistically significant contribution of education and political identification was 12%. Similar to the prediction of the connection factor, participants with higher education level ($B = -.09$, $SE = .03$, $p = .004$) and who self-identified more as Republican ($B = -.16$, $SE = .02$, $p < .001$) rated the integrity factor of authenticity statistically significantly lower.

## Discussion

This research developed and tested a novel tool for measuring authenticity in the context of science communication that can shape and inform trust and credibility research. This research also discovered that if a scientist shares the origin story of their research in first-person, people are more inclined to perceive the scientist as authentic. If a scientist uses first-person alone in the scientific brief, people are more inclined to perceive the scientist as authentic based on a feeling of connection. These findings offer both theoretical and practical headway in the understanding of authenticity in science communication.

Dimensions of perceived authenticity in response to the narrative message conditions strongly aligned with the qualities of trustworthiness identified in the literature as benevolence and integrity. More specifically, the message testing suggested that authenticity in a science communication context is connected more strongly to benevolence than integrity. However, the results of our factor analysis also offered an intriguing suggestion that the established concept of benevolence might be expanded to include perceptions of the researcher's passion for

their own work and the potential relevance of that work to the audience. We consequently characterized the benevolence category more broadly as connection—to oneself, to one's work, and to one's audience.

Because the factor categorized in our study as connection captures not just the warmth and goodwill that the scientist may feel for the audience, but also warmth and goodwill that is generated toward the scientist in return, it emphasizes the reciprocal nature of the communication exchange and establishment of trustworthiness. It addresses such questions as: Does the researcher care about me understanding their research, does the researcher care whether or not the research is relevant to my life, is the researcher willing to show me their true self in a way that invites my connection to them, and even empathy? This reciprocity may be particularly crucial when the power (i.e. knowledge/expertise) differences are more pronounced and/or the conditions exist for motivated reasoning (i.e. the researcher exhibits sociocultural cues that are at odds with the audience's). While acknowledging that the connection measure could be argued to represent diverse components (i.e., the relationship of the researcher with the audience, with their work, and with their self-understanding), our factor analysis indicates that there is a valid reason for grouping these items together. By identifying a link between the researcher's attitude toward the work/audience and the audience's emotional response to the researcher, this study raises interesting new questions. If, as Petraglia claims [25], "authenticity is not exclusively—or even predominantly—about the objective accuracy of information as much as it is about one's attitude toward information," it is not just the audience's attitude toward the information that is in play in assessing the potential contribution of perceived authenticity in trustworthiness, but the researcher's as well. This research points toward an expansion of the idea of benevolence to include these additional characteristics.

Among the four narrative conditions this study tested—each based on qualities that the literature suggested were associated with perceived authenticity—only the narrative condition that combined the first-person narrative style with the researcher's origin of interest was rated as significantly more authentic for the connection and integrity factors than the control. Just using a first-person narrative style resulted in perceptions of more authenticity on the connection factor, which was also rated significantly higher than the control. The message conditions designed to illustrate vulnerability in the scientist (backward-looking fallibility) and openness about scientific uncertainty (forward-looking fallibility) did not result in increased perceptions of authenticity that were statistically significant. The challenge of measuring the impact of scientific uncertainty on perceptions of authenticity points to a limitation of this study. Scientific uncertainty is itself multi-faceted and under-defined in the literature [50]. Further, uncertainty is a primary driver of scientific research generally and findings are rarely presented without some acknowledgement of how knowledge on the subject is evolving. Thus, by underscoring the openness of the communicator toward revision—both past and future—message conditions 3 and 5 amplify a kind of uncertainty that is nonetheless also present in the control message to a lesser degree.

Another limitation suggested by this result relates to the relative difficulty of drafting narrative conditions that reflect complex and/or subjective attributes such as vulnerability and openness, as compared to the comparatively concrete narrative task of describing how a scientist came to be interested in the subject at hand. All experimental message conditions were designed to be of the same approximate length, ranging from 175 words (message condition 5) to 194 words (message condition 4), with the exception of message condition 2 which involved a change to the first person but otherwise included no new information. Dramatizing forward- and backward-looking fallibility by adding concrete examples might well have made them more similar to message condition 4 in vividness, but would have likely required a trade-off with regard to similarity in length. This dilemma underscores a challenge of narrative research

generally and represents a limitation in this study: narratives are heterogeneous and multi-dimensional and the preponderance of interdisciplinary definitions of narrative focus on structure rather than content. While some recent research offers a framework for examining the power of some narratives to engage over others [51], the absence of useful methods to compare narrative "power" represents a gap in the study of narratives and warrants further study.

At the same time, the fact that the first-person narrative style combined with presenting the researcher's origin of interest in the scientific brief did produce significant results indicates this might be an effective technique for conveying the researcher's "personal stake in the issue" [24] in a manner that promotes a sense of authenticity.

Though research suggests that some individuals may be more inclined to trust others in general (Mayer et al., 1995), the literature further suggests that a variety of social factors may predispose certain groups to trust or not trust scientific leaders and that these factors may vary significantly according to context [1]. Political ideology, education level, race/ethnicity, and gender have all been shown to have impact in some settings and with some topics [32, 36, 52–53]. In particular, the literature suggests that political affiliation and education level impact trust in science communication messages in politicized contexts [31, 54–55]. Findings in this study discovered some of those same predispositions, as evidenced by differences in perceived authenticity related to education level and political identification that emerged in a non-politicized context.

Because politicized contexts have been shown to influence trust, the authors intentionally chose a topic—the domestication of plants—that has not been politicized for the purpose of measuring perceived authenticity in response to narrative manipulations. Neither education nor political affiliation were predicted to exert significant effects on the perception of authenticity in response to our message conditions. However, regardless of the message condition, the participants with higher educational level and who self-identified more as Republican rated statistically significantly lower scores for both the connection and integrity factors of authenticity. Although it is possible that the topic of plant domestication has political dimensions of which the researchers were unaware, it is more likely that these results reflect political and educational predispositions toward scientists in general.

## Practical applications and future research

In conceptualizing and measuring perceived authenticity in science communication, this research speaks to a gap highlighted in the NAS report: "Given the importance of audience perceptions about the trustworthiness and credibility of the communicator, research needs to examine the effects on audiences when science communicators are open about their own values and preferences" [1]. The discovery that by sharing the origin of his/her interest in a subject a scientist might enhance perceived authenticity is a strategy that is importantly not dependent upon the audience's understanding or acceptance of other indicators, such as institutional affiliation.

While, as mentioned above, the conceptual framework of connection will benefit from future study, this research nonetheless hints at the ways that perceived authenticity may differ from other causes of trust such as transparency or honesty. A science communicator who is transparent and honest can be expected to reveal their institutional ties. Institutional trust, however, is more influenced by the kind of sociocultural factors at work in motivated reasoning than trust in an individual person [14] and so, to the extent that critiques of scientists focus on the argument that they are self-interested, biased or possessing a hidden agenda [54], the attempt by the scientist to communicate scientific objectivity and institutional authority may work against their trustworthiness. The objective style of traditional science communication—

meant to convey "abstract truths that remain valid" regardless of context [41]—is disadvantaged in a media environment that privileges personal narratives from a variety of professional and non-professional sources [3]. By refocusing on the individual communicator, however briefly, these narrative techniques have the potential to create opportunities for connection that might be otherwise closed off.

The Internet provides ample opportunity for consumers to apply sociocultural criteria to the filtering of information, including scientific information. In this context, conventional hallmarks of expertise such as advanced degrees, institutional affiliation, and claims of objectivity [15] may provoke motivated reasoning related to sociocultural factors, as opposed to conferring authority, as the science communicator may hope. This presents science communicators with the following dilemma, particularly when they are communicating findings in politicized contexts: A scientist must simultaneously present themselves as *unlike* their audience in an important way (they have information to share that is the result of rigorous research—they are an expert with important findings) while also convincing the audience that they are not *so* unlike them (politically, or in their specific professional and personal goals) that they should be dismissed as untrustworthy. This dilemma becomes more pronounced the more specialized and complicated the findings and the more necessary the reliance on expertise [56]. As conceptualized by this research, authenticity offers an opportunity for scientist and audience to recognize each other as individuals with qualities in common that, while expressed in the context of science communication, are not dependent upon the science itself (passion, eagerness to be understood, a specific personal history).

In the current media and political environment, many influences on how audiences apply sociocultural filters in forming perceptions of a scientist's ability/expertise are beyond the scientist's control. Understanding how a scientist might increase the degree of trust by strengthening credibility with regard to one specific component may allow them to offset other potential deficits. The narrative manipulation tested in this first study highlights an approach —communicating the scientist's origin of interest in the topic—that is concrete and available to a given science communicator, regardless of the external environment. The next phase of research will study whether or not conveying a sense of the scientist as a human being, a unique individual with qualities *beyond* institutional affiliations or a role in the production of the research, has the power to mediate motivated reasoning with regard to the communication of politicized findings.

## Supporting information

**S1 Appendix. Messages used in each experimental condition and a full list of survey questions.**
(DOCX)

## Acknowledgments

We gratefully acknowledge Dr. J. Chris Pires from the Division of Biological Sciences at the University of Missouri for his professional and generous consultation on the scientific materials used in our experiment.

## Author Contributions

**Conceptualization:** Lise Saffran, Amanda Hinnant.

**Data curation:** Sisi Hu.

**Formal analysis:** Sisi Hu.

**Funding acquisition:** Lise Saffran.

**Investigation:** Lise Saffran, Amanda Hinnant.

**Methodology:** Sisi Hu, Laura D. Scherer.

**Project administration:** Lise Saffran.

**Resources:** Lise Saffran.

**Supervision:** Lise Saffran, Amanda Hinnant.

**Validation:** Laura D. Scherer.

**Writing – original draft:** Lise Saffran, Sisi Hu, Amanda Hinnant, Susan C. Nagel.

**Writing – review & editing:** Lise Saffran, Sisi Hu, Amanda Hinnant, Laura D. Scherer, Susan C. Nagel.

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
