## [Decision Letter · Decision Letter 0]

10 Sep 2019

PONE-D-19-16714

Constructing and influencing perceived authenticity in science communication: Experimenting with narrative

PLOS ONE

Dear Ms. Hu,

Thank you for submitting your manuscript to PLOS ONE. After careful consideration, we feel that it has merit but does not fully meet PLOS ONE’s publication criteria as it currently stands. Therefore, we invite you to submit a revised version of the manuscript that addresses the points raised during the review process.

We would appreciate receiving your revised manuscript by Oct 25 2019 11:59PM. To enhance the reproducibility of your results, we recommend that if applicable you deposit your laboratory protocols in protocols.io, where a protocol can be assigned its own identifier (DOI) such that it can be cited independently in the future. For instructions see: http://journals.plos.org/plosone/s/submission-guidelines#loc-laboratory-protocols

We look forward to receiving your revised manuscript.

Kind regards,

Dion R. J. O'Neale, Ph.D.

Academic Editor

PLOS ONE

Additional Editor Comments:

I apologise for the slow decision on this. I had hoped to receive responses from two reviewers, however was twice let down by reviewers who failed to return reports after agreeing to review the manuscript.

The referee's report below identifies some points where the MS could be strengthened and improved by, mostly through discussion or clarification. While these have been stated succinctly, they do constitute reasonably significant additions to the MS and I would like to see them address substantively before publication.

Reviewers' comments:

Reviewer's Responses to Questions

**Comments to the Author**

1. Is the manuscript technically sound, and do the data support the conclusions?

Reviewer #1: Partly

2. Has the statistical analysis been performed appropriately and rigorously? 

Reviewer #1: I Don't Know

3. Have the authors made all data underlying the findings in their manuscript fully available?

Reviewer #1: Yes

4. Is the manuscript presented in an intelligible fashion and written in standard English?

Reviewer #1: Yes

5. Review Comments to the Author

Reviewer #1: The authors have done an excellent job of situating their study in the context of the existing literature in an emerging area that has direct relevance for science communication practitioners. I particularly appreciate their focus on advancing knowledge along lines of enquiry highlighted in the National Academies of Science 2017 research agenda (referenced throughout).

The study design avoids excessive complication and is explained clearly and accessibly in the article (with the exception of the Factor analysis through to the Exploratory analyses sections, which would benefit from some further revisions to improve readability).

However, a close reading raises questions about the influence of the specific text samples used to represent the four experimental conditions. Is it possible that variations in the quality of the writing may have influenced the results in a way that is currently unaccounted for in the authors' analysis? For example, the text used to measure condition 4 (referencing the origin of the science communicator’s interest in the subject under study) references travelling in space and alien life forms. It contains the lines, "Later, studying plants, I realized that they had all the variety and strangeness I had loved as a child. Studying plants was almost like studying aliens." This stands out as a much more intriguing, colourful and memorable statement that the variations present in the other text samples used in the experiment, which are notably more abstract and impersonal despite being written in first-person narrative style (e.g. "my earlier conclusions focused too much on human decisions and not enough on accidental events").

Condition 4 provided the most significant result of the study, according to the authors' analysis, and they conclude that referencing the origin of the science communicator's interest in the topic increases their perceived authenticity. However, it seems worthwhile to question whether this conclusion is actually supported by the data, or whether additional variables (e.g. salience, emotional resonance) introduced through variations in the text samples are responsible.

Another aspect to note regarding the text samples is that the attempt to isolate factors measuring the effect of including uncertainty (condition 3) and referencing past mistakes (condition 5) is potentially undermined by the inclusion of the lines "Until recently, scientists/we thought..." and "...earlier than [I had] previously thought" in all conditions, including the control. In my view, these lines already introduce some degree of uncertainty and show a willingness to update previous errors in interpretation for all the conditions evaluated. This means that the authors' conclusion that these factors were less important for scientists' perceived authenticity cannot be supported by the data as presented. I suggest revisions to this portion of the discussion.

Taking a further step back, the decision to include first-person narrative as a baseline for all four experimental conditions, rather than independently examining the influence of factors like acknowledging uncertainty and referencing past mistakes on perceptions of scientists' authenticity is not sufficiently clear. There is certainly interest among science communication practitioners in whether these factors have a role to play in strengthening perceptions of authenticity. Since the hypothesis about first person narrative style (H1) was only partially supported by the study, this raises questions about whether the other experimental factors could have been more effective in enhancing perceived authenticity on their own (e.g. a third-person statement that references the origin of scientists' general interest in a subject).

In regards to methods, I note that the authors do not explain why they chose to include reverse wording versions of a subset of the survey questions, despite the fact that the usefulness of this practice is frequently called into question. (See, for example:

https://journals.plos.org/plosone/article?id=10.1371/journal.pone.0157795

https://journals.plos.org/plosone/article?id=10.1371/journal.pone.0068967 )

Moreover, it is unclear why only reverse worded questions passed the statistical threshold for inclusion under the factor labelled "integrity" (as reported in the data statement). More explanation is needed to show whether the inclusion of reverse wording has introduced an additional factor that may have confounded the analysis.

To aid with proofreading, a few errors and further observations:

Line 260, "Reverse wording"? or should this actually read "reverse coding" to agree with the data statement

Line 284, delete "that"

Line 314, suggest structuring Table 1 to rank items listed under each factor by the strength of their loading (highest to lowest).

Line 326, spell out acronyms when first used

Line 462, what is meant by "the diversity in the connection measure"?

Line 507, missing full stop (period)

Data made available: It would be helpful to include the full list of survey questions and full text of the conditions in the same location.

6. PLOS authors have the option to publish the peer review history of their article (what does this mean?). If published, this will include your full peer review and any attached files.

Reviewer #1: Yes: Dacia Herbulock

---

## [Author Response · Author response to Decision Letter 0]

17 Oct 2019

Dear Dr. O’Neale,

What follows is our response to the reviewer’s comments, all of which we addressed below in italics. We thank the reviewer, Dacia Herbulock, for her time and expertise. 

Reviewer’s comments:

The authors have done an excellent job of situating their study in the context of the existing literature in an emerging area that has direct relevance for science communication practitioners. I particularly appreciate their focus on advancing knowledge along lines of enquiry highlighted in the National Academies of Science 2017 research agenda (referenced throughout).

Thank you for this positive feedback.

The study design avoids excessive complication and is explained clearly and accessibly in the article (with the exception of the Factor analysis through to the Exploratory analyses sections, which would benefit from some further revisions to improve readability).

We revised to make those sections to be more readable.

However, a close reading raises questions about the influence of the specific text samples used to represent the four experimental conditions. Is it possible that variations in the quality of the writing may have influenced the results in a way that is currently unaccounted for in the authors’ analysis? For example, the text used to measure condition 4 (referencing the origin of the science communicator’s interest in the subject under study) references travelling in space and alien life forms. It contains the lines, “Later, studying plants, I realized that they had all the variety and strangeness I had loved as a child. Studying plants was almost like studying aliens.” This stands out as a much more intriguing, colourful and memorable statement that the variations present in the other text samples used in the experiment, which are notably more abstract and impersonal despite being written in first-person narrative style (e.g. “my earlier conclusions focused too much on human decisions and not enough on accidental events”).

Condition 4 provided the most significant result of the study, according to the authors’ analysis, and they conclude that referencing the origin of the science communicator’s interest in the topic increases their perceived authenticity. However, it seems worthwhile to question whether this conclusion is actually supported by the data, or whether additional variables (e.g. salience, emotional resonance) introduced through variations in the text samples are responsible.

The two comments above, both rooted in the reviewer’s concern that the influence of the messages may be biased by the variations in the quality of the writing, are well taken. We made a significant revision to the discussion on page 24 to address this concern. The explanation includes an acknowledgment that most narrative research focuses on structure and that few frameworks currently exist to compare narratives for their power to engage based on content (cited work of Shaffer et al. 2018). In this study, structural measures to create comparable narratives included an effort to keep the narratives as close as possible to the same length, while including enough content to convey the primary differences in the message conditions (all experimental conditions were within twenty words of each other). This exposes another limitation of the study that is directly tied to the qualities of authenticity highlighted in the message conditions: some (e.g. presenting unproven or contradictory findings) are based on more abstract concepts than others (e.g. origin of researcher’s interest). This relates to the structural concerns above, in that concepts that are themselves concrete can be conveyed economically. Conveying abstract concepts would likely require additional text that would include both an explanation of the abstract concept (e.g. I made a mistake) and a concrete example as an illustration.

Another aspect to note regarding the text samples is that the attempt to isolate factors measuring the effect of including uncertainty (condition 3) and referencing past mistakes (condition 5) is potentially undermined by the inclusion of the lines “Until recently, scientists/we thought...” and “...earlier than [I had] previously thought” in all conditions, including the control. In my view, these lines already introduce some degree of uncertainty and show a willingness to update previous errors in interpretation for all the conditions evaluated. This means that the authors’ conclusion that these factors were less important for scientists’ perceived authenticity cannot be supported by the data as presented. I suggest revisions to this portion of the discussion.

The reviewer correctly identifies the possibility that some level of uncertainty exists in the message conditions, even before we introduced uncertainty and referenced past mistakes in message conditions three and five. The goal of this research was to compare manipulated messages with conventional modes of scientific communication. We acknowledge the fact that the communication of scientific findings is rarely presented without some indication of the context of the research, which necessarily locates these findings within an evolving understanding of the subject. We also acknowledge the difficulty in isolating uncertainty here given that various levels of uncertainty are conventionally represented in scientific communication and that scientific inquiry is itself driven in large part by uncertainty (and cite Han et al. on pages 23 and 24 in the discussion). Consequently, messages three and five were designed to amplify qualities of vulnerability (willingness to admit mistakes) and openness to potentially contradictory data in the future (uncertainty). In recognition of the reviewer insight that some level of uncertainty exists throughout, we have renamed message conditions three and five to better express the kind of uncertainty in focus. Beginning with the “Hypothesis and Research Question” section and continuing throughout the paper, message condition 3 was formerly “first person combined with uncertainty in the findings” and has now been renamed “findings in first person with forward-looking fallibility.” Message condition 5 was formerly “admitting an earlier/possible misconception” and has now been renamed “presenting backward-looking fallibility.” 

Taking a further step back, the decision to include first-person narrative as a baseline for all four experimental conditions, rather than independently examining the influence of factors like acknowledging uncertainty and referencing past mistakes on perceptions of scientists’ authenticity is not sufficiently clear. There is certainly interest among science communication practitioners in whether these factors have a role to play in strengthening perceptions of authenticity. Since the hypothesis about first person narrative style (H1) was only partially supported by the study, this raises questions about whether the other experimental factors could have been more effective in enhancing perceived authenticity on their own (e.g. a third-person statement that references the origin of scientists’ general interest in a subject).

Thank you for pointing this out. We added a rationale for the use of first-person in all experimental conditions (not control) on page 9. Presenting qualities such as vulnerability in the third person introduces yet another person into the communication process--the narrator. In that way it would risk undermining the connection between the message and communicator.

In regards to methods, I note that the authors do not explain why they chose to include reverse wording versions of a subset of the survey questions, despite the fact that the usefulness of this practice is frequently called into question. (See, for example:

https://journals.plos.org/plosone/article?id=10.1371/journal.pone.0157795

https://journals.plos.org/plosone/article?id=10.1371/journal.pone.0068967)

Moreover, it is unclear why only reverse worded questions passed the statistical threshold for inclusion under the factor labelled “integrity” (as reported in the data statement). More explanation is needed to show whether the inclusion of reverse wording has introduced an additional factor that may have confounded the analysis.

Thank you for pointing out the issues regarding reverse coding. We added a rationale (see pages 12 and 13) for the use of reverse-worded questions. We acknowledge that using reverse wording is ineffective to reduce response bias being introduced by participants’ acquiescence, inattention, or confusion (cited from the second article you provided, Van Sonderen et al., 2013); however, using reverse worded questions in this study is not for those purposes, but for reducing the difficulty in interpreting the questions. Those seven questions sounded more natural and therefore easier to understand when phrasing in a reverse direction. Thus, we decided to keep those reverse-worded questions in the questionnaire and reverse-coded them in the data analysis. 

We agree with the reviewer that it seems suspicious that only reverse worded questions loaded under the factor of integrity. To address this concern, we added statistical evidence (i.e., correlation between these two factors) and citations (Harrington, 2009) to show connection and integrity are distinct factors (see page 15). 

To aid with proofreading, a few errors and further observations:

Thank you. We revised each of them.

Line 260, “Reverse wording”? or should this actually read “reverse coding” to agree with the data statement

We switched it to “reverse coding” (now on line 298).

Line 284, delete “that”

We deleted “that” (now on line 329).

Line 314, suggest structuring Table 1 to rank items listed under each factor by the strength of their loading (highest to lowest).

We restructured Table 1 to rank items by the strength of their loadings from highest to lowest.

Line 326, spell out acronyms when first used

We spelled out the full name of MANCOVA, multivariate analysis of covariance (now on line 391).

Line 462, what is meant by “the diversity in the connection measure”?

Thank you for point this out. We reworded this sentence and added specific meaning of diversity in the brackets (see page 23). 

Line 507, missing full stop (period)

We added a period there.

Data made available: It would be helpful to include the full list of survey questions and full text of the conditions in the same location.

Thank you for your suggestion. We added the full list of survey questions into S1 Appendix, together with the full text of the conditions.

Thanks to the reviewer again for the beneficial suggestions.

Sincerely,

(Name blinded)

References in response to reviewers:

Han PKJ, Klein WMP, Arora NK. Varieties of Uncertainty in Health Care: A Conceptual Taxonomy. Med Decis Making. 2011;31(6):828-38.

Harrington D. Confirmatory factor analysis. New York, NY: Oxford university press; 2009.

Osborne J, Costello A, Kellow J. Best practices in exploratory factor analysis. 2008 2019/10/16. In: Best Practices in Quantitative Methods [Internet]. Thousand Oaks, CA: SAGE Publications, Inc. Available from: https://methods.sagepub.com/book/best-practices-in-quantitative-methods.

Shaffer VA, Focella ES, Hathaway A, Scherer LD, Zikmund-Fisher BJ. On the Usefulness of Narratives: An Interdisciplinary Review and Theoretical Model. Ann Behav Med. 2018;52(5):429-42.

Sonderen Ev, Sanderman R, Coyne JC. Ineffectiveness of Reverse Wording of Questionnaire Items: Let’s Learn from Cows in the Rain. PLoS ONE. 2013;8(7):e68967.

---

## [Decision Letter · Decision Letter 1]

5 Dec 2019

Constructing and influencing perceived authenticity in science communication: Experimenting with narrative

PONE-D-19-16714R1

Dear Dr. Hu,

We are pleased to inform you that your manuscript has been judged scientifically suitable for publication and will be formally accepted for publication once it complies with all outstanding technical requirements.

With kind regards,

Dion R. J. O'Neale, Ph.D.

Academic Editor

PLOS ONE

Additional Editor Comments (optional):

Reviewers' comments:

Reviewer's Responses to Questions

**Comments to the Author**

1. If the authors have adequately addressed your comments raised in a previous round of review and you feel that this manuscript is now acceptable for publication, you may indicate that here to bypass the “Comments to the Author” section, enter your conflict of interest statement in the “Confidential to Editor” section, and submit your "Accept" recommendation.

Reviewer #1: All comments have been addressed

2. Is the manuscript technically sound, and do the data support the conclusions?

Reviewer #1: (No Response)

3. Has the statistical analysis been performed appropriately and rigorously? 

Reviewer #1: (No Response)

4. Have the authors made all data underlying the findings in their manuscript fully available?

Reviewer #1: (No Response)

5. Is the manuscript presented in an intelligible fashion and written in standard English?

Reviewer #1: (No Response)

6. Review Comments to the Author

Reviewer #1: (No Response)

7. PLOS authors have the option to publish the peer review history of their article (what does this mean?). If published, this will include your full peer review and any attached files.

Reviewer #1: Yes: Dacia Herbulock

---

## [Editor Report · Acceptance letter]

17 Dec 2019

PONE-D-19-16714R1 

Constructing and influencing perceived authenticity in science communication: Experimenting with narrative 

Dear Dr. Hu:

I am pleased to inform you that your manuscript has been deemed suitable for publication in PLOS ONE. Congratulations! Your manuscript is now with our production department. 

With kind regards,

on behalf of

Dr. Dion R. J. O'Neale 

Academic Editor

PLOS ONE